# Value Propagation for Decentralized Networked Deep Multi-agent Reinforcement Learning

Chao Qu [*1], Shie Mannor[2], Huan Xu[3,4], Yuan Qi[1], Le Song[1,4], and Junwu Xiong[1]

[1]Ant Financial Services Group
[2] Technion
[3]Alibaba Group
[4]Georgia Institute of Technology

## Abstract

We consider the networked multi-agent reinforcement learning (MARL) problem in a fully decentralized setting, where agents learn to coordinate to achieve joint success. This problem is widely encountered in many areas including traffic control, distributed control, and smart grids. We assume each agent is located at a node of a communication network and can exchange information only with its neighbors. Using softmax temporal consistency, we derive a primal-dual decentralized optimization method and obtain a principled and data-efficient iterative algorithm named *value propagation*. We prove a non-asymptotic convergence rate of $\mathcal{O}(1/T)$ with nonlinear function approximation. To the best of our knowledge, it is the first MARL algorithm with a convergence guarantee in the control, off-policy, non-linear function approximation, fully decentralized setting.

## 1  Introduction

Multi-agent systems have applications in a wide range of areas such as robotics, traffic control, distributed control, telecommunications, and economics. For these areas, it is often difficult or simply impossible to predefine agents' behaviour to achieve satisfactory results, and multi-agent reinforcement learning (MARL) naturally arises [Bu et al., 2008, Tan, 1993]. For example, El-Tantawy et al. [2013] model a traffic signal control problem as a multi-player stochastic game and solve it with MARL. MARL generalizes reinforcement learning by considering a set of agents (decision makers) sharing a common environment. However, multi-agent reinforcement learning is a challenging problem since the agents interact with both the environment and each other. For instance, independent Q-learning—treating other agents as a part of the environment—often fails as the multi-agent setting breaks the theoretical convergence guarantee of Q-learning and makes the learning process unstable [Tan, 1993]. Rashid et al. [2018], Foerster et al. [2018], Lowe et al. [2017] alleviate such a problem using a centralized network (i.e., being centralized for training, but decentralized during execution.). Its communication pattern is illustrated in the left panel of Figure 1.

Despite the great success of (partially) centralized MARL approaches, there are various scenarios, such as sensor networks [Rabbat and Nowak, 2004] and intelligent transportation systems [Adler and Blue, 2002] , where a central agent does not exist or may be too expensive to use. In addition, *privacy* and *security* are requirements of many real world problems in multi-agent system (also in many modern machine learning problems) [Abadi et al., 2016, Kurakin et al., 2016] . For instance, in Federated learning [McMahan et al., 2016], the learning task is solved by a lose federation of participating devices (agents) without the need to centrally store the data, which significantly reduces

---

[*]luoji.qc@antfin.com

privacy and security risk by limiting the attack surface to only the device. In the agreement problem [DeGroot, 1974, Mo and Murray, 2017], a group of agents may want to reach consensus on a subject without leaking their *individual goal* or *opinion* to others. Obviously, centralized MARL violates privacy and security requirements. To this end, we and others have advocated the *fully decentralized* approaches, which are useful for many applications including unmanned vehicles [Fax and Murray, 2002], power grid [Callaway and Hiskens, 2011], and sensor networks [Cortes et al., 2004]. For these approaches, we can use a network to model the interactions between agents (see the right panel of Figure 1). Particularly, We consider a *fully cooperative* setting where each agent makes its own decision based on its *local reward* and messages received from their neighbors. Thus each agent preserves the *privacy* of its own *goal* and *policy*. At the same time, through the message-passing all agents achieve consensus to maximize the averaged cumulative rewards over all agents; see Equation (3).

In this paper, we propose a new fully decentralized networked multi-agent deep reinforcement learning algorithm. Using softmax temporal consistency [Nachum et al., 2017, Dai et al., 2018] to connect value and policy updates, we derive a new two-step primal-dual decentralized reinforcement learning algorithm inspired by a primal decentralized optimization method [Hong et al., 2017] [2]. In the first step of each iteration, each agent computes its local policy, value gradients and dual gradients and then updates only policy parameters. In the second step, each agent propagates to its neighbors the messages based on its value function (and dual function) and then updates its own value function. Hence we name the algorithm *value propagation*. It preserves the privacy in the sense that no indi-

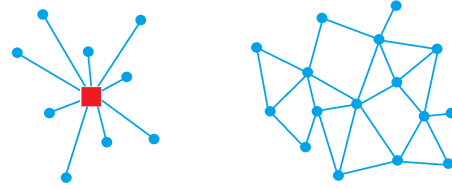

Figure 1: Centralized network vs Decentralized network. Each blue node in the figure corresponds to an agent. In centralized network (left), the red central node collects information for all agents, while in decentralized network (right), agents exchanges information with neighbors.

vidual reward function is required for the network-wide collaboration. We approximate the local policy, value function and dual function of each agent by deep neural networks, which enables automatic feature generation and end-to-end learning.

**Contributions:** [1] We propose the value propagation algorithm and prove that it converges with the rate $\mathcal{O}(1/T)$ even with the *non-linear* deep neural network function approximation. To the best of our knowledge, it is the first deep MARL algorithm with non-asymptotic convergence guarantee. At the same time, value propagation can use off-policy updates, making it data efficient. When it reduces to the single agent case, it provides a proof of [Dai et al., 2018] in the realistic setting; see remarks of algorithm 1 in Section 3.3. [2] The objective function in our problem is a primal-dual decentralized optimization form (see (8)), while the objective function in [Hong et al., 2017] is a primal problem. When our method reduces to pure primal analysis, we extend [Hong et al., 2017] to the stochastic and biased gradient setting which may be of independent interest to the optimization community. In the practical implementation, we extend ADAM into the decentralized setting to accelerate training.

## 2   Preliminaries

**MDP**  Markov Decision Process (MDP) can be described by a 5-tuple $(\mathcal{S}, \mathcal{A}, \mathcal{R}, \mathcal{P}, \gamma)$: $\mathcal{S}$ is the finite state space, $\mathcal{A}$ is the finite action space, $\mathcal{P} = (P(s'|s, a))_{s,s'\in\mathcal{S},a\in\mathcal{A}}$ are the transition probabilities, $R = (R(s, a))_{s,s'\in\mathcal{S},a\in\mathcal{A}}$ are the real-valued immediate rewards and $\gamma \in (0, 1)$ is the discount factor. A policy is used to select actions in the MDP. In general, the policy is stochastic and denoted by $\pi$, where $\pi(s_t, a_t)$ is the conditional probability density at $a_t$ associated with the policy. Define $V^*(s) = \max_\pi \mathbb{E}[\sum_{t=0}^{\infty} \gamma^t R(s_t, a_t)|s_0 = s]$ to be the optimal value function. It is known that $V^*$ is the unique fixed point of the Bellman optimality operator, $V(s) = (\mathcal{T}V)(s) := \max_a R(s, a) + \gamma\mathbb{E}_{s'|s,a}[V(s')]$. The optimal policy $\pi^*$ is related to $V^*$ by the following equation: $\pi^*(s, a) = \arg\max_a\{R(s, a) + \gamma\mathbb{E}_{s'|s,a}V^*(s')\}$

**Softmax Temporal Consistency** Nachum et al. [2017] establish a connection between value and policy based reinforcement learning based on a relationship between softmax temporal value consistency and policy optimality under entropy regularization. Particularly, the soft Bellman optimality is as follows,

$$V_\lambda(s) = \max_{\pi(s,\cdot)} \big(\mathbb{E}_{a\sim\pi(s,\cdot)}(R(s,a) + \gamma\mathbb{E}_{s'|s,a}V_\lambda(s')) + \lambda H(\pi,s)\big), \tag{1}$$

where $H(\pi,s) = -\sum_{a\in\mathcal{A}} \pi(s,a)\log\pi(s,a)$ and $\lambda \geq 0$ controls the degree of regularization. When $\lambda = 0$, above equation reduces to the standard Bellman optimality condition. An important property of soft Bellman optimality is the called temporal consistency, which leads to the Path Consistency Learning.

**Proposition 1.** *[Nachum et al., 2017]. Assume $\lambda > 0$. Let $V_\lambda^*$ be the fixed point of* (1) *and $\pi_\lambda^*$ be the corresponding policy that attains that maximum on the RHS of* (1)*. Then, $(V_\lambda^*, \pi_\lambda^*)$ is the unique $(V,\pi)$ pair that satisfies the following equation for all $(s,a) \in \mathcal{S} \times \mathcal{A} : V(s) = R(s,a) + \gamma\mathbb{E}_{s'|s,a}V(s') - \lambda\log\pi(s,a)$.*

A straightforward way to apply temporal consistency is to optimize the following problem, $\min_{V,\pi} E_{s,a}\big(R(s,a) + \gamma\mathbb{E}_{s'|s,a}V(s') - \lambda\log\pi(s,a) - V(s)\big)^2$. Dai et al. [2018] get around the double sampling problem of above formulation by introduce a primal-dual form

$$\min_{V,\pi} \max_\rho \mathbb{E}_{s,a,s'}[(\delta(s,a,s') - V(s))^2] - \eta\mathbb{E}_{s,a,s'}[(\delta(s,a,s') - \rho(s,a))^2], \tag{2}$$

where $\delta(s,a,s') = R(s,a) + \gamma V(s') - \lambda\log\pi(s,a)$, $0 \leq \eta \leq 1$ controls the trade-off between bias and variance.

In the following discussion, we use $\|\cdot\|$ to denote the Euclidean norm over the vector, $A'$ stands for the transpose of $A$, and $\odot$ denotes the entry-wise product between two vectors.

# 3 Value Propagation

In this section, we present our multi-agent reinforcement learning algorithm, i.e., value propagation. To begin with, we extend the MDP model to the Networked Multi-agent MDP model following the definition in [Zhang et al., 2018]. Let $\mathcal{G} = (\mathcal{N}, \mathcal{E})$ be an undirected graph with $|\mathcal{N}| = N$ agents (node). $\mathcal{E}$ represents the set of edges. $(i,j) \in \mathcal{E}$ means agent $i$ and $j$ can communicate with each other through this edge. A networked multi-agent MDP is characterized by a tuple $(\mathcal{S}, \{\mathcal{A}^i\}_{i\in\mathcal{N}}, \mathcal{P}, \{R^i\}_{i\in\mathcal{N}}, \mathcal{G}, \gamma)$: $\mathcal{S}$ is the global state space shared by all agents (It could be partially observed, i.e., each agent observes its own state $S^i$, see our experiment). $\mathcal{A}^i$ is the action space of agent $i$, $\mathcal{A} = \prod_{i=1}^N \mathcal{A}^i$ is the joint action space, $\mathcal{P}$ is the transition probability, $\mathcal{R}^i$ denotes the *local* reward function of agent $i$. We assume rewards are observed only locally to preserve the privacy of the each agent's goal. At each time step, agents observe $s_t$ and make the decision $a^t = (a_1^t, a_2^t, ..., a_N^t)$. Then each agent just receives its own reward $R_i(s_t, a_t)$, and the environment switches to the new state $s_{t+1}$ according to the transition probability. Furthermore, since each agent make the decisions independently, it is reasonable to assume that the policy $\pi(s,a)$ can be factorized, i.e., $\pi(s,a) = \prod_{i=1}^N \pi^i(s,a^i)$ [Zhang et al., 2018]. We call our method *fully-decentralized* method, since reward is received locally, the action is executed locally by agent, critic (value function) are trained locally.

## 3.1 Multi-Agent Softmax Temporal Consistency

The goal of the agents is to learn a policy that maximizes the long-term reward averaged over the agent, i.e.,

$$\mathbb{E}\sum_{t=0}^\infty \frac{1}{N}\sum_{i=1}^N \gamma^t R_i(s_t, a_t). \tag{3}$$

In the following, we adapt the temporal consistency into the multi-agent version. Let $V_\lambda(s) = \mathbb{E}\big(\frac{1}{N}\sum_{i=1}^N R_i(s,a) + \gamma\mathbb{E}_{s'|s,a}V_\lambda(s') + \lambda H(\pi,s)\big)$, $V_\lambda^*$ be the optimal value function and $\pi_\lambda^*(s,a) = \prod_{i=1}^N \pi_\lambda^{i*}(s,a^i)$ be the corresponding policy. Apply the soft temporal consistency, we obtain that for

all $(s, a) \in \mathcal{S} \times \mathcal{A}$, $(V_\lambda^*, \pi_\lambda^*)$ is the unique $(V, \pi)$ pair that satisfies

$$V(s) = \frac{1}{N} \sum_{i=1}^{N} R_i(s, a) + \gamma \mathbb{E}_{s'|s,a} V(s') - \lambda \sum_{i=1}^{N} \log \pi^i(s, a^i). \tag{4}$$

A optimization problem inspired by (4) would be

$$\min_{\{\pi^i\}_{i=1}^N, V} \mathbb{E}\big(V(s) - \frac{1}{N} \sum_{i=1}^{N} R_i(s, a) - \gamma \mathbb{E}_{s'|s,a} V(s') + \lambda \sum_{i=1}^{N} \log \pi^i(s, a^i)\big)^2. \tag{5}$$

There are two potential issues of above formulation: First, due to the inner conditional expectation, it would require two independent samples to obtain the unbiased estimation of gradient of $V$ [Dann et al., 2014]. Second, $V(s)$ is a global variable over the network, thus can not be updated in a *decentralized* way.

For the first issue, we introduce the primal-dual form of (5) as that in [Dai et al., 2018]. Using the fact that $x^2 = \max_\nu (2\nu x - \nu^2)$ and the interchangeability principle [Shapiro et al., 2009] we have,

$$\min_{V, \{\pi^i\}_{i=1}^N} \max_\nu 2\mathbb{E}_{s,a,s'}[\nu(s,a)\big(\frac{1}{N} \sum_{i=1}^{N} (R_i(s,a) + \gamma V(s') - V(s) - \lambda N \log \pi^i(s,a^i))\big)] - \mathbb{E}_{s,a,s}[\nu^2(s,a)].$$

Change the variable $\nu(s, a) = \rho(s, a) - V(s)$, the objective function becomes

$$\min_{V, \{\pi^i\}_{i=1}^N} \max_\rho \mathbb{E}_{s,a,s'}[\big(\frac{1}{N} \sum_{i=1}^{N} (\delta_i(s, a, s') - V(s))\big)^2] - \mathbb{E}_{s,a,s'}[\big(\frac{1}{N} \sum_{i=1}^{N} (\delta_i(s, a, s') - \rho(s, a))\big)^2], \tag{6}$$

where $\delta_i = R_i(s, a) + \gamma V(s') - \lambda N \log \pi^i(s, a^i)$.

## 3.2 Decentralized Formulation

So far the problem is still in a centralized form, and we now turn to reformulating it in a decentralized way. We assume that policy, value function, dual variable $\rho$ are all in the parametric function class. Particularly, each agent's policy is $\pi^i(s, a^i) := \pi_{\theta_{\pi i}}(s, a^i)$ and $\pi_\theta(s, a) = \prod_{i=1}^{N} \pi_{\theta_{\pi i}}(s, a^i)$. The value function $V_{\theta_v}(s)$ is characterized by the parameter $\theta_v$, while $\theta_\rho$ represents the parameter of $\rho(s, a)$. Similar to [Dai et al., 2018], we optimize a slightly different version from (6).

$$\min_{\theta_v, \{\theta_{\pi i}\}_{i=1}^N} \max_{\theta_\rho} \mathbb{E}_{s,a,s'}[\big(\frac{1}{N} \sum_{i=1}^{N} (\delta_i(s, a, s') - V(s))\big)^2] - \eta \mathbb{E}_{s,a,s'}[\big(\frac{1}{N} \sum_{i=1}^{N} (\delta_i(s, a, s') - \rho(s, a))\big)^2], \tag{7}$$

where $0 \le \eta \le 1$ controls the bias and variance trade-off. When $\eta = 0$, it reduces to the pure primal form.

We now consider the second issue that $V(s)$ is a global variable. To address this problem, we introduce the local copy of the value function, i.e., $V_i(s)$ for each agent $i$. In the algorithm, we have a consensus update step, such that these local copies are the same, i.e., $V_1(s) = V_2(s) = ... = V_N(s) = V(s)$, or equivalently $\theta_{v^1} = \theta_{v^2} = ... = \theta_{v^N}$, where $\theta_{v^i}$ are parameter of $V_i$ respectively. Notice now in (7), there is a global dual variable $\rho$ in the primal-dual form. Therefore, we also introduce the local copy of the dual variable, i.e., $\rho_i(s, a)$ to formulate it into the decentralized optimization problem. Now the *final* objective function we need to optimize is

$$\min_{\{\theta_{v^i}, \theta_{\pi^i}\}_{i=1}^N} \max_{\{\theta_{\rho^i}\}_{i=1}^N} L(\theta_V, \theta_\pi, \theta_\rho) = \mathbb{E}_{s,a,s'}[\big(\frac{1}{N} \sum_{i=1}^{N} (\delta_i(s, a, s') - V_i(s))\big)^2]$$

$$- \eta \mathbb{E}_{s,a,s'}[\big(\frac{1}{N} \sum_{i=1}^{N} (\delta_i(s, a, s') - \rho_i(s, a))\big)^2],$$

$$s.t. \ \theta_{v^1} =, ..., = \theta_{v^N}, \theta_{\rho^1} =, ..., = \theta_{\rho^N}, \tag{8}$$

where $\delta_i = R_i(s, a) + \gamma V_i(s') - \lambda N \log \pi^i(s, a^i)$. We are now ready to present the value propagation algorithm. In the following, for notational simplicity, we assume the parameter of each agent is a

scalar, i.e., $\theta_{\rho^i}, \theta_{\pi^i}, \theta_{v^i} \in R$. We pack the parameter together and slightly abuse the notation by writing $\theta_\rho = [\theta_{\rho^1}, ..., \theta_{\rho^N}]'$, $\theta_\pi = [\theta_{\pi^1}, ..., \theta_{\pi^N}]'$, $\theta_V = [\theta_{v^1}, ..., \theta_{v^N}]'$. Similarly, we also pack the stochastic gradient $g(\theta_\rho) = [g(\theta_{\rho^1}), ..., g(\theta_{\rho^n})]'$, $g(\theta_V) = [g(\theta_{v^1}), ..., g(\theta_{v^n})]'$.

### 3.3 Value propagation algorithm

Solving (8) even without constraints is not an easy problem when both primal and dual parts are approximated by the deep neural networks. An ideal way is to optimize the inner dual problem and find the solution $\theta_\rho^* = \arg\max_{\theta_\rho} L(\theta_V, \theta_\pi, \theta_\rho)$, such that $\theta_{\rho^1} = ... = \theta_{\rho^N}$. Then we can do the (decentralized) stochastic gradient decent to solve the primal problem.

$$\min_{\{\theta_{v^i}, \theta_{\pi^i}\}_{i=1}^N} L(\theta_V, \theta_\pi, \theta_\rho^*) \ \ s.t. \ \ \theta_{v^1} = ... = \theta_{v^N}. \tag{9}$$

However in practice, one *tricky issue* is that we can not get the exact solution $\theta_\rho^*$ of the dual problem. Thus, we do the (decentralized) stochastic gradient for $T_{dual}$ steps in the dual problem and get an approximated solution $\tilde{\theta}_\rho$ in the Algorithm 1. In our analysis, we take the error $\varepsilon$ generated from this inexact solution into the consideration and analyze its effect on the convergence. Particularly, since $\nabla_{\theta_V} L(\theta_V, \theta_\pi, \tilde{\theta}_\rho) \neq \nabla_{\theta_V} L(\theta_V, \theta_\pi, \theta_\rho^*)$, the primal gradient is biased and the results in [Dai et al., 2018, Hong et al., 2017] do not fit this problem.

In the dual update we do a consensus update $\theta_\rho^{t+1} = \frac{1}{2} D^{-1} L^+ \theta_\rho^t - \frac{\alpha_\rho}{2} D^{-1} A' \mu_\rho^t + \frac{\alpha_\rho}{2} D^{-1} g(\theta_\rho^t)$ using the stochastic gradient of each agent, where $\mu_\rho$ is some auxiliary variable to incorporate the communication, $D$ is the degree matrix, $A$ is the node-edge incidence matrix, $L^+$ is sign-less graph Laplacian. We defer the detail definition and the derivation of this algorithm to Appendix A.1 and Appendix A.5 due to space limitation. After updating the dual parameters, we optimize the primal parameters $\theta_v, \theta_\pi$. Similarly, we use a mini-batch data from the replay buffer and then do a consensus update on $\theta_v$. The same remarks on $\rho$ also hold for the primal parameter $\theta_v$. Notice here we do not need the consensus update on $\theta_\pi$, since each agent's policy $\pi^i(s, a^i)$ is different than each other. This update rule is adapted from a primal decentralized optimization algorithm [Hong et al., 2017]. Notice even in the pure primal case, Hong et al. [2017] only consider the batch gradient case while our algorithm and analysis include the stochastic and biased gradient case. In practicals implementation, we consider the decentralized momentum method and multi-step temporal consistency to accelerate the training; see details in Appendix A.2 and Appendix A.3.

**Remarks on Algorithm 1**. **(1)** In the *single* agent case, Dai et al. [2018] assume the dual problem can be exactly solved and thus they analyze a simple pure primal problem. However such assumption is unrealistic especially when the dual variable is represented by the deep neural network. Our *multi-agent* analysis considers the *inexact* solution. This is much harder than that in [Dai et al., 2018], since now the primal gradient is biased. **(2)** The update of each agent just needs the information of the agent itself and its neighbors. See this from the definition of $D$, $A$, $L^+$ in the appendix. **(3)** The topology of the Graph $\mathcal{G}$ affects the convergence speed. In particular, the rate depends on $\sigma_{\min}(A'A)$ and $\sigma_{\min}(D)$, which are related to spectral gap of the network.

## 4 Theoretical Result

In this section, we give the convergence result on Algorithm 1. We first make two mild assumptions on the function approximators $f(\theta)$ of $V_i(s), \pi^i(s, a^i), \rho_i(s, a)$.

**Assumption 1.** i) The function approximator $f(\theta)$ is differentiable and has Lipschitz continuous gradient, i.e., $\|\nabla f(\theta_1) - \nabla f(\theta_2)\| \leq L\|\theta_1 - \theta_2\|, \forall \theta_1, \theta_2 \in R^K$. This is commonly assumed in the non-convex optimization. ii) The function approximator $f(\theta)$ is lower bounded. This can be easily satisfied when the parameter is bounded, i.e., $\|\theta\| \leq C$ for some positive constant $C$.

In the following, we give the theoretical analysis for Algorithm 1 in the same setting of [Antos et al., 2008, Dai et al., 2018] where samples are prefixed and from one single $\beta$-mixing off-policy sample path. We denote $\hat{L}(\theta_V, \theta_\pi) = \max_{\theta_\rho} L(\theta_V, \theta_\pi, \theta_\rho), s.t., \theta_{\rho^1} =, ..., = \theta_{\rho^N}$

**Theorem 1.** *Let the function approximators of $V_i(s)$, $\pi^i(s, a^i)$ and $\rho_i(s, a)$ satisfy Assumption 1, snd denote the total training step be $T$. We solve the inner dual problem with a approximated solution $\tilde{\theta}_\rho = (\tilde{\theta}_{\rho^1}, ..., \tilde{\theta}_{\rho^N})'$, such that $\|\nabla_{\theta_V} L(\theta_V, \theta_\pi, \tilde{\theta}_\rho) - \nabla_{\theta_V} L(\theta_V, \theta_\pi, \theta_\rho^*)\| \leq c_1/\sqrt{T}$, and $\|\nabla_{\theta_\pi} L(\theta_V, \theta_\pi, \tilde{\theta}_\rho) - \nabla_{\theta_\pi} L(\theta_V, \theta_\pi, \theta_\rho^*)\| \leq c_2/\sqrt{T}$. Assume the variance of the stochastic gradient*

---
**Algorithm 1** Value Propagation
---

Input: Environment ENV, learning rate $\alpha_\pi, \alpha_v, \alpha_\rho$, discount factor $\gamma$, number of step $T_{dual}$ to train dual parameter $\theta_{\rho^i}$, replay buffer capacity $B$, node-edge incidence matrix $A \in R^{E \times N}$, degree matrix $D$, signless graph Laplacian $L^+$.

Initialization of $\theta_{v^i}, \theta_{\pi^i}, \theta_{\rho^i}, \mu_\rho^0 = 0, \mu_V^0 = 0$.

**for** $t = 1, ..., T$ **do**

    sample trajectory $s_{0:\tau} \sim \pi(s, a) = \prod_{i=1}^N \pi^i(s, a^i)$ and add it into the replay buffer.

    **1. Update the dual parameter** $\theta_{\rho^i}$

    Do following dual update $T_{dual}$ times:

    Random sample a mini-batch of transition $(s_t, \{a_t^i\}_{i=1}^N, s_{t+1}, \{r_t^i\}_{i=1}^N)$ from the replay buffer.

    **for** agent $i = 1$ to $n$ **do**

        Calculate the stochastic gradient $g(\theta_{\rho^i}^t)$ of $-\eta(\delta_i(s_t, a_t, s_{t+1}) - \rho_i(s_t, a_t))^2$ w.r.t. $\theta_{\rho^i}^t$.

    **end for**

    // Do consensus update on $\theta_\rho := [\theta_{\rho^1}, ..., \theta_{\rho^N}]'$

    $\theta_\rho^{t+1} = \frac{1}{2}D^{-1}L^+\theta_\rho^t - \frac{\alpha_\rho}{2}D^{-1}A'\mu_\rho^t + \frac{\alpha_\rho}{2}D^{-1}g(\theta_\rho^t), \mu_\rho^{t+1} = \mu_\rho^t + \frac{1}{\alpha_\rho}A\theta_\rho^{t+1}$

    **2. Update primal parameters** $\theta_{v^i}, \theta_{\pi^i}$

    Random sample a mini-batch of transition $(s_t, \{a_t^i\}_{i=1}^N, s_{t+1}, \{r_t^i\}_{i=1}^N)$ from the replay buffer.

    **for** agent $i = 1$ to $n$ **do**

        Calculate the stochastic gradient $g(\theta_{v^i}^t), g(\theta_{\pi^i}^t)$ of $(\delta_i(s_t, a_t, s_{t+1}) - V_i(s_t))^2 - \eta(\delta_i(s_t, a_t, s_{t+1}) - \rho_i(s_t, a_t))^2$, w.r.t. $\theta_{v^i}^t, \theta_{\pi^i}^t$

    **end for**

    // Do gradient decent on $\theta_{\pi^i}$: $\theta_{\pi^i}^{t+1} = \theta_{\pi^i}^t - \alpha_\pi g(\theta_{\pi^i}^t)$ for each agent $i$.

    // Do consensus update on $\theta_V := [\theta_{v^1}, ..., \theta_{v^N}]'$ :

    $\theta_V^{t+1} = \frac{1}{2}D^{-1}L^+\theta_V^t - \frac{\alpha_v}{2}D^{-1}A'\mu_V^t - \frac{\alpha_v}{2}D^{-1}g(\theta_V^t), \mu_V^{t+1} = \mu_V^t + \frac{1}{\alpha_v}A\theta_V^{t+1}$.

**end for**

---

*$g(\theta_V)$, $g(\theta_\pi)$ and $g(\theta_\rho)$ (estimated by a single sample) are bounded by $\sigma^2$, the size of the mini-batch is $\sqrt{T}$, the step size $\alpha_\pi, \alpha_v, \alpha_\rho \propto \frac{1}{L}$. Then value propagation in Algorithm 1 converges to the stationary solution of $\hat{L}(\theta_V, \theta_\pi)$ with rate $\mathcal{O}(1/T)$.*

**Remarks: (1)** The convergence criteria and its dependence on the network structure are involved. We defer the definition of them to the proof section in the appendix (Equation (44)). **(2)** We require that the approximated dual solution $\tilde{\theta}_\rho$ are not far from $\theta_\rho^*$ such that the estimation of the primal gradient of $\theta_v$ and $\theta_\pi$ are not far from the true one (the distance is less than $\mathcal{O}(1/\sqrt{T})$). Once the inner dual problem is concave, we can get this approximated solution easily using vanilla decentralized stochastic gradient method after at most $T$ steps. If the dual problem is non-convex, we still can show the dual problem converges to some stationary solution with rate $\mathcal{O}(1/T)$ by our proof. **(3)** In the theoretical analysis, the stochastic gradient estimated from the mini-batch (rather than the estimation from a single sample ) is common in non-convex analysis, see the work [Ghadimi and Lan, 2016]. In practice, a mini-batch of samples is commonly used in training deep neural network.

## 5 Related work

Among related work on MARL, the setting of [Zhang et al., 2018] is close to ours, where the authors proposed a fully decentralized multi-agent Actor-Critic algorithm to maximize the expected time-average reward $\lim_{T\to\infty} \frac{1}{T}\mathbb{E}\sum_{t=1}^T \frac{1}{n}\sum_{i=1}^n r_i^t$. They provide the *asymptotic* convergence analysis on the *on-policy* and *linear* function approximation setting. In our work, we consider the discounted reward setup, i.e., Equation (3). Our algorithm includes both on-policy and *off-policy* setting thus can exploit data more efficiently. Furthermore, we provide a convergence rate $\mathcal{O}(\frac{1}{T})$ in the *non-linear* function approximation setting which is much stronger than the result in [Zhang et al., 2018]. Littman [1994] proposed the framework of Markov games which can be applied to collaborative and competitive setting [Lauer and Riedmiller, 2000, Hu and Wellman, 2003]. These early works considered the tabular case thus can not apply to real problems with large state space. Recent works [Foerster et al., 2016, 2018, Rashid et al., 2018, Raileanu et al., 2018, Jiang et al., 2018, Lowe et al., 2017] have exploited powerful deep learning and obtained some promising empirical results. However most of them lacks theoretical guarantees while our work provides convergence analysis.

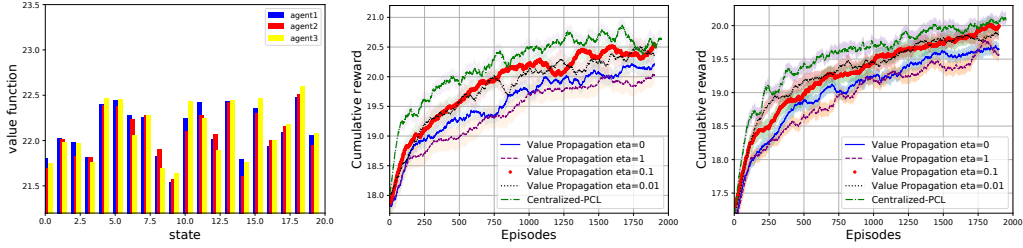

Figure 2: Results on randomly sampled MDP. Left: Value function of different agents in value propagation. In the figure, value functions of three agents are similar, which means agents get consensus on value functions. Middle: Cumulative reward of value propagation (with different $\eta$) and centralized PCL with 10 agents. Right : Results with 20 agents.

We emphasize that most of the research on MARL is in the fashion of centralized training and decentralized execution. In the training, they do not have the constraint on the communication, while our work has a network decentralized structure.

# 6    Experimental result

The goal of our experiment is two-fold: To better understand the effect of each component in the proposed algorithm; and to evaluate efficiency of value propagation in the off-policy setting. To this end, we first do an ablation study on a simple random MDP problem, we then evaluate the performance on the cooperative navigation task [Lowe et al., 2017]. The settings of the experiment are similar to those in [Zhang et al., 2018]. Some implementation details are deferred to Appendix A.4 due to space constraints.

## 6.1    Ablation Study

In this experiment, we test effect of several components of our algorithm such as the consensus update, dual formulation in a random MDP problem. Particularly we answer following three questions: **(1)** Whether an agent can get consensus through message-passing in value propagation even when each agent just knows its local reward. **(2)** How much performance does the decentralized approach sacrifice comparing with centralized one? **(3)** What is the effect of the dual part in our formulation ($0 \le \eta \le 1$ and $\eta = 0$ corresponds to the pure primal form)?

We compare value propagation with the *centralized* PCL. The centralized PCL means that there is a central node to collect rewards of all agent, thus it can optimize the objective function (5) using the single agent PCL algorithm [Nachum et al., 2017, Dai et al., 2018]. Ideally, value propagation should converges to the same long term reward with the one achieved by the centralized PCL. In the experiment, we consider a multi-agent RL problem with $N = 10$ and $N = 20$ agents, where each agent has two actions. A discrete MDP is randomly generated with $|\mathcal{S}| = 32$ states. The transition probabilities are distributed uniformly with a small additive constant to ensure ergodicity of the MDP, which is $\mathcal{P}(s'|a, s) \propto p_{ss'}^a + 10^{-5}, p_{ss'}^a \sim U[0, 1]$. For each agent $i$ and each state-action pair $(s, a)$, the reward $R_i(s, a)$ is uniformly sampled from $[0, 4]$.

In the left panel of Figure 2, we verify that the value function $v_i(s)$ in value propagation reaches the consensus through message-passing in the end of the training. Particularly, we randomly choose three agent $i$, $j$, $k$ and draw their value functions over 20 randomly picked states. It is easy to see that value functions $v_i(s), v_j(s), v_k(s)$ over these states are almost same. This is accomplished by the consensus update in value propagation. In the middle and right panel of Figure 2, we compare the result of value propagation with centralized PCL and evaluate the effect of the dual part of value propagation. Particularly, we pick $\eta = 0, 0.01, 0.1, 1$ in the experiment, where $\eta = 0$ corresponds to the pure primal formulation. When $\eta$ is too large ($\eta = 1$), the algorithm would have large variance while $\eta = 0$ the algorithm has some bias. Thus value propagation with $\eta = 0.1, 0.01$ has better result. We also see that value propagation ($\eta = 0.1, 0.01$) and centralized PCL converge to almost the same value, although there is a gap between centralized and decentralized algorithm. The centralized PCL converges faster than value propagation, since it does not need time to diffuse the reward information over the network.

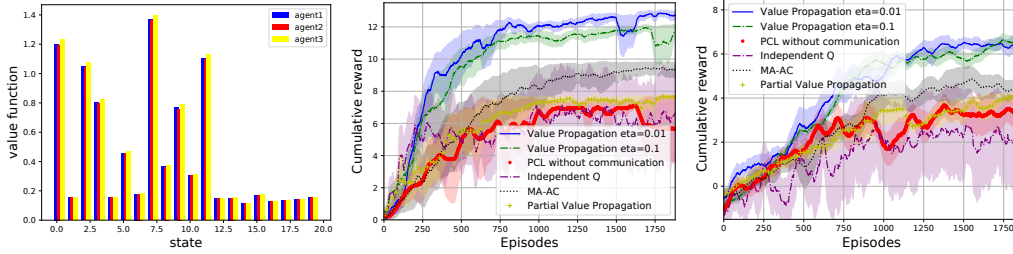

Figure 3: Results on Cooperative Navigation task. Left: value functions of three random picked agents (totally 16 agents) in value propagation. They get consensus. Middle : cumulative reward of value propagation (eta=0.01 and eta=0.1), MA-AC and PCL without communication with agent number N=8. Right: Results with agent number N=16. Our algorithm outperforms MA-AC and PCL without communication. Comparing with the middle panel, the number of agent increases in the right panel. Therefore, the problem becomes harder (more collisions). We see agents achieve lower cumulative reward (averaged over agents) and need more time to find a good policy.

## 6.2 Cooperative Navigation task

The aim of this section is to demonstrate that the value propagation outperforms decentralized multi-agent Actor-Critic (MA-AC)[Zhang et al., 2018], independent Q learning [Tan, 1993], the Multi-agent PCL *without communication*. Here PCL without communication means each agent maintains its own estimation of policy $\pi^i(s, a^i)$ and value function $V^i(s)$ but there is no communication Graph. Notice that this is different from the centralized PCL in Section 6.1, where centralized PCL has a central node to collect all reward information and thus do not need further communication. Note that the original MA-AC is designed for the averaged reward setting thus we adapt it into the discounted case to fit our setting. We test the value propagation in the environment of the Cooperative Navigation task [Lowe et al., 2017], where agents need to reach a set of $L$ landmarks through physical movement. We modify this environment to fit our setting. A reward is given when the agent reaches its own landmarks. A penalty is received if agents collide with other agents. Since the position of landmarks are different, the reward function of each agent is different. Here we test the case the state is globally observed and partially observed. In particular, we assume the environment is in a rectangular region with size $2 \times 2$. There are $N = 8$ or $N = 16$ agents. Each agent has a single target landmark, i.e., $L = N$, which is randomly located in the region. Each agent has five actions which corresponds to going up, down, left, right with units 0.1 or staying at the position. The agent has high probability (0.95) to move in the direction following its action and go in other direction randomly otherwise. The maximum length of each epoch is set to be 500 steps. When the agent is close enough to the landmark, e.g., the distance is less than 0.1, we think it reaches the target and gets reward $+5$. When two agents are close to each other (with distance less than 0.1), we treat this case as a collision and a penalty $-1$ is received for each of the agents. The state includes the position of the agents. The communication graph is generated as that in Section 6.1 with connectivity ratio $4/N$. In the partially observed case, the actor of each agent can only observe its own and neighbors' states. We report the results in Figure 3.

In the left panel of Figure 3, we see the value function $v_i(s)$ reaches consensus in value propagation. In the middle and right panel of Figure 3, we compare value propagation with PCL without communication, independent Q learning and MA-AC. In PCL without communication, each agent maintains its own policy, value function and dual function, which is trained by the algorithm SBEED [Dai et al., 2018] with $\eta = 0.01$. Since there is no communication between agents, intuitively agents may have more collisions in the learning process than those in value propagation. Similar augment holds for the independent Q learning. Indeed, In the middle and right panel, we see value propagation learns the policy much faster than PCL without communication. We also observe that value propagation outperforms MA-AC. One possible reason is that value propagation is an off-policy method thus we can apply experience replay which exploits data more efficiently than the on-policy method MA-AC. We also test the performance of value propagation (result labeled as partial value propagation in Figure 3) when the state information of actor is partially observed. Since the agent has limited information, its performance is worse than the fully observed case. But it is better than the PCL without communication (fully observed state).

## Footnotes

[2]The objective in Hong et al. [2017] is a primal optimization problem with constraint. Thus they introduce a Lagrange multiplier like method to solve it (so they call it primal-dual method ). Our objective function is a primal-dual optimization problem with constraint.

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
