[Supplementary Material]

## A

### A.1 Topology of the Graph

Here, we explain the matrix in the Algorithm 1 which are closely related to the topology of the Graph, which is left from the main paper due to the limit of the space.

- $D = diag[d_1, ..., d_N]$ is the degree matrix, with $d_i$ denoting the degree of node $i$.
- $A$ is the node-edge incidence matrix: if $e \in \mathcal{E}$ and it connects vertex i and j with $i > j$, then $A_{ev} = 1$ if $v = i$, $A_{ev} = -1$ if $v = j$ and $A_{ev} = 0$ otherwise.
- The signless incidence matrix $B := |A|$, where the absolute value is taken for each component of $A$.
- The signless graph Laplacian $L^+ = B^T B$. By definition $L^+(i, j) = 0$ if $(i, j) \notin \mathcal{E}$. Notice the non-zeros element in $A$, $L^+$, the update just depends on each agent itself and its neighbor.

### A.2 Practical Acceleration

The algorithm 1 trains the agent with vanilla gradient decent method with a extra consensus update. In practice, the adaptive momentum gradient methods including Adagrad Duchi et al. [2011], Rmsprop Tieleman and Hinton and Adam Kingma and Ba [2014] have much better performance in training the deep neural network. We adapt Adam in our setting, and propose algorithm 2 which has better performance than algorithm 1 in practice.

---

**Algorithm 2** Accelerated value propagation

---

Input: Environment ENV, learning rate $\beta_1, \beta_2 \in [0, 1)$, $\alpha_t$, discount factor $\gamma$, a mixing matrix $W$, number of step $T_{dual}$ to train dual parameter $\theta_\rho^i$, replay buffer capacity $B$.

Initialization of $\theta_{v^i}, \theta_{\pi^i}, \theta_{\rho^i}$, moment vectors $m_{v^i}^0 = m_{\rho^i}^0 = 0$, $w_{v^i}^0 = w_{\rho^i}^0 = 0$ .

**for** $t = 1, ..., T$ **do**

    sample trajectory $s_{0:\tau} \sim \pi(s, a) = \prod_{i=1}^N \pi^i(s, a^i)$ and add it into the replay buffer.
    // **Update the dual parameter** $\theta_{\rho^i}$
    Do following update $T_{dual}$ times:
    Random sample a mini-batch of transition $(s_t, \{a_t^i\}_{i=1}^N, s_{t+1}, \{r_t^i\}_{i=1}^N)$ from the replay buffer.
    **for** agent $i = 1$ to $n$ **do**
        Calculate the stochastic gradient $g(\theta_{\rho^i}^t)$ of $-\eta(\delta_i(s_t, a_t, s_{t+1}) - \rho_i(s_t, a_t))^2$ w.r.t. $\theta_{\rho^i}^t$.
        // update momentum parameters: $m_{\rho^i}^t = \beta_1 m_{\rho^i}^{t-1} + (1 - \beta_1)(-g(\theta_{\rho^i}^t))$
        $w_{\rho^i}^t = \beta_2 w_{\rho^i}^{t-1} + (1 - \beta_2)g(\theta_{\rho^i}^t) \odot g(\theta_{\rho^i}^t)$
    **end for**
    // Do consensus update for each agent $i$
    $\theta_{\rho^i}^{t+\frac{1}{2}} = \sum_{j=1}^N [W]_{ij} \theta_{\rho^j}^t$, $\theta_{\rho^i}^t = \theta_{\rho^i}^{t+\frac{1}{2}} - \alpha_t \frac{m_{\rho^i}^t}{\sqrt{w_{\rho^i}^t}}$
    // **End the update of dual problem**
    // **Update primal parameters** $\theta_{v^i}, \theta_{\pi^i}$.
    Random sample a mini-batch of transition $(s_t, \{a_t^i\}_{i=1}^N, s_{t+1}, \{r_t^i\}_{i=1}^N)$ from the replay buffer.
    **for** agent $i = 1$ to $n$ **do**
        Calculate the stochastic gradient $g(\theta_{v^i}^t)$,$g(\theta_{\pi^i}^t)$ of $(\delta_i(s_t, a_t, s_{t+1}) - V_i(s_t))^2 - \eta(\delta_i(s_t, a_t, s_{t+1}) - \rho_i(s_t, a_t))^2$, w.r.t. $\theta_{v^i}^t, \theta_{\pi^i}^t$
        //update the momentum parameter:
        $m_{v^i}^t = \beta_1 m_{v^i}^{t-1} + (1 - \beta_1)g(\theta_{v^i}^t)$
        $w_{v^i}^t = \beta_2 w_{v^i}^{t-1} + (1 - \beta_2)g(\theta_{v^i}^t) \odot g(\theta_{v^i}^t)$
        // Using Adam to update $\theta_{\pi^i}$ for each agent $i$.
        // Do consensus update on $\theta_{v^i}$ for each agent $i$:
        $\theta_{v^i}^{t+\frac{1}{2}} = \sum_{j=1}^N [W]_{ij} \theta_{v^j}^t$, $\theta_{v^i}^t = \theta_{v^i}^{t+\frac{1}{2}} - \alpha_t \frac{m_{v^i}^t}{\sqrt{w_{v^i}^t}}$
    **end for**
**end for**

---

**Mixing Matrix:** In Algorithm 2, there is a mixing matrix $W \subset R^{N \times N}$ in the consensus update. As its name suggests, it mixes information of the agent and its neighbors. This nonnegative matrix $W$ need to satisfy the following condition.

- $W$ needs to be doubly stochastic, i.e., $W^T \mathbf{1} = \mathbf{1}$ and $W\mathbf{1} = \mathbf{1}$.

- $W$ respects the communication graph $\mathcal{G}$, i.e., $W(i, j) = 0$ if $(i, j) \notin \mathcal{E}$.

- The spectral norm of $W^T(I - \mathbf{1}\mathbf{1}^T/N)W$ is strictly smaller than one.

Here is one particular choice of the mixing matrix $W$ used in our work which satisfies above requirement called Metropolis weights Xiao et al. [2005].

$$W(i, j) = 1 + \max[d(i), d(j)]^{-1}, \forall (i, j) \in \mathcal{E},$$
$$W(i, i) = 1 - \sum_{j \in NE(i)} W(i, j), \forall i \in \mathcal{N}, \tag{10}$$

where $NE(i) = \{j \in \mathcal{N} : (i, j) \in \mathcal{E}\}$ is the set of neighbors of the agent $i$ and $d(i) = |\mathcal{N}(i)|$ is the degree of agent $i$. Such mixing matrix is widely used in decentralized and distributed optimization Boyd et al. [2006], Cattivelli et al. [2008]. The update rule of the momentum term in Algorithm 2 is adapted from Adam. The consensus (communication) steps are $\theta_{\rho^i}^{t+\frac{1}{2}} = \sum_{j=1}^{N}[W]_{ij}\theta_{\rho^j}^t$ and $\theta_{v^i}^{t+\frac{1}{2}} = \sum_{j=1}^{N}[W]_{ij}\theta_{v^j}^t$.

### A.3 Multi-step Extension on value propagation

The temporal consistency can be extended to the multi-step case Nachum et al. [2017], where the following equation holds

$$V(s_0) = \sum_{t=0}^{k-1} \gamma^t \mathbb{E}_{s_t|s_0, a_{0:t-1}}[R(s_t, a_t) - \lambda \log \pi(s_t, a_t^i)] + \gamma^k \mathbb{E}_{s_k|s_0, a_{0:k-1}} V(s_k).$$

Thus in the objective function (8), we can replace $\delta_i$ by $\delta_i(s_{0:k}, a_{0:k-1}) = \sum_{t=0}^{k-1} \gamma^t (R_i(s_t, a_t) - \lambda N \log \pi^i(s_t, a_t^i)) + \gamma^k V_i(s_k)$ and change the estimation of stochastic gradient correspondingly in Algorithm 1 and Algorithm 2 to get the multi-step version of vaue propagation . In practice, the performance of setting $k > 1$ is better than $k = 1$ which is also observed in single agent case Nachum et al. [2017], Dai et al. [2018]. We can tune $k$ for each application to get the best performance.

### A.4 Implementation details of the experiments

**Ablation Study**

The value function $v_i(s)$ and dual variable $\rho_i(s, a)$ are approximated by two hidden-layer neural network with Relu as the activation function where each hidden-layer has 20 hidden units. The policy of each agent is approximated by a one hidden-layer neural network with Relu as the activation function where the number of the hidden units is 32. The output is the softmax function to approximate $\pi^i(s, a^i)$. The mixing matrix in Algorithm 2 is selected as the Metropolis Weights in (10). The graph $\mathcal{G}$ is generated by randomly placing communication links among agents such that the connectivity ratio is $4/N$. We set $\gamma = 0.9$, $\lambda = 0.01$, learning rate $\alpha$=5e-4. The choice of $\beta_1$, $\beta_2$ are the default value in Adam.

**Cooperative Navigation task**

The value function $v_i(s)$ is approximated by a two-hidden-layer neural network with Relu as the activation function where inputs are the state information. Each hidden-layer has 40 hidden units. The dual function $\rho(s, a)$ is also approximated by a two-hidden-layer neural network, where the only difference is that inputs are state-action pairs (s,a). The policy is approximated by a one-hidden-layer neural network with Relu as the activation function. The number of the hidden units is 32. The output is the softmax function to approximate $\pi^i(s, a^i)$. In all experiments, we use the multi-step version of value propagation and choose $k = 4$. We choose $\gamma = 0.95$, $\lambda = 0.01$. The learning rate of Adam is chosen as 5e-4 and $\beta_1$, $\beta_2$ are default value in Adam optimizer. The setting of PCL

 without communication is exactly same with value propagation except the absence of communication
460 network.

## A.5 Consensus update in Algorithm 1

462 We now give details to derive the Consensus Update in Algorithm 1 with $\eta = 1$ to ease the exposition.
463 When $\eta \in [0, 1)$, we just need to change variable and some notations, the result are almost same.
464 Here we use the primal update as an example, the derivation of the dual update is the same.

465 In the main paper section 3, we have shown that when $\eta = 1$, in the primal update, we basically solve
466 following problem.

$$\min_{\{\theta_{v_i}, \theta_{\pi_i}\}_{i=1}^N} 2\mathbb{E}_{s,a,s}[\nu^*(s,a)\big(\frac{1}{N}\sum_{i=1}^N (R^i(s,a) + \gamma V_i(s') - V_i(s) - \lambda N \log \pi^i(s,a))\big)] - \mathbb{E}_{s,a,s}[\nu^{*2}(s,a)],$$
$$s.t., \theta_{v_1} = ... = \theta_{v_n}.$$
$$(11)$$

467 here for simplicity we assume in the dual optimization, we have already find the optimal solution
468 $\nu^*(s,a)$. It can be any approximated solution of $\tilde{\nu}(s,a)$ which does not affect the derivation of the
469 update rule in primal optimization. In the later proof, we will show how this approximated solution
470 affects the convergence rate.

471 When we optimize w.r.t. $\theta_{v^i}$, we basically we solve a non-convex problem with the following form

$$\min_x f(x) = \sum_{i=1}^N f_i(x_i), \;\; s.t. \;\; x_1 = ... = x_N \qquad (12)$$

472 Recall the definition of the node-edge incidence matrix $A$: if $e \in \mathcal{E}$ and it connects vertex $i$ and $j$
473 with $i > j$, then $A_{ev} = 1$ if $v = i$, $A_{ev} = -1$ if $v = j$ and $A_e v = 0$ otherwise. Thus by define
474 $x = [x_1, ..., x_N]'$ we have a equivalent form of (12)

$$\min_x f(x) = \sum_{i=1}^N f_i(x_i), \;\; s.t., \;\; Ax = 0 \qquad (13)$$

475 Notice the update of $\theta_{\pi^i}$ is a special case of above formulation, since we do not have the constraint
476 $x_1 =, ..., = x_N$. Thus in the following, it suffice to analyze above formulation (13). We adapt the
477 Prox-PDA in Hong et al. [2017] to solve above problem. To keep the notation consistent with Hong
478 et al. [2017], we consider a more general problem

$$\min_x f(x) = \sum_{i=1}^N f_i(x_i), s.t., Ax = b.$$

479 In the following we denote $\nabla f(x^t) := [(\nabla_{x_1} f(x_1))^T, ..., (\nabla_{x_N} f(x_N))^T]^T$ where the superscript $'$
480 means transpose. We denote $g_i(x_i)$ as an estimator of $\nabla_{x_i} f(x_i)$ and $g(x) = [g_1(x_1), ..., g_N(x_N)]$.

481 The update rule of Prox-PDA is

$$x^{t+1} = \arg\min_x \langle g(x^t), x - x^t \rangle + \langle \mu^t, Ax - b \rangle + \frac{\beta}{2}\|Ax - b\|^2 + \frac{\beta}{2}\|x - x^t\|_{B^T B}^2 \qquad (14)$$

$$\mu^{t+1} = \mu^t + \beta(Ax^{t+1} - b) \qquad (15)$$

482 where $g(x^t)$ is an estimator of $\nabla f(x^t)$. The signed graph Laplacian matrix $L_-$ is $A^T A$. Now we
483 choose $B := |A|$ as the signless incidence matrix. Using this choice of $B$, we have $B^T B = L^+ \in$
484 $\mathbb{R}^{N \times N}$ which is the signless graph Laplacian whose $(i, i)$th diagonal entry is the degree of node $i$
485 and its $(i, j)$th entry is 1 if $e = (i, j) \in \mathcal{E}$, and 0 otherwise.

Thus

$$x^{t+1} = \arg\min_x \langle g(x^t), x - x^t \rangle + \langle \mu^t, Ax - b \rangle + \frac{\beta}{2} x^T L_- x + \frac{\beta}{2} (x - x^t) L^+ (x - x^t)$$

$$= \arg\min_x \langle g(x^t), x \rangle + \langle \mu^t, Ax - b \rangle + \frac{\beta}{2} x^T (L_- + L^+) x - \beta x^T L^+ x^t \qquad (16)$$

$$= \arg\min_x \langle g(x^t), x \rangle + \langle \mu^t, Ax - b \rangle + \beta x^T D x - \beta x^T L^+ x^t,$$

where $D = diag[d_1, ..., d_N]$ is the degree matrix, with $d_i$ denoting the degree of node $i$.

After simple algebra, we obtain

$$x^{t+1} = \frac{1}{2} D^{-1} L^+ x^t - \frac{1}{2\beta} D^{-1} A^T \mu^t - \frac{1}{2\beta} D^{-1} g(x^t),$$

which is the primal update rule of the consensus step in the algorithm 1 (notice here the stepsize is $1/\beta$)

# B  Convergence Proof of Value Propagation

## B.1  Convergence on the primal update

In this section, we first give the convergence analysis of the value propagation (algorithm 1) on the primal update. To include the effected of the inexact solution of dual optimization problem, we denote $g(x^t) = \nabla f(x^t) + \epsilon_t$, where $\epsilon_t = \varepsilon_t + \tilde{\varepsilon}_t$ is some error terms.

- $\varepsilon_t$ is a zero mean random variable coming from the randomness of the stochastic gradient $g(x^t)$.
- $\tilde{\varepsilon}_t$ comes from the approximated solution of $\tilde{\nu}$ in (11) or $\tilde{\rho}$ in (8) such that $\|\nabla_{\theta_v} L(\theta_V, \theta_\pi, \tilde{\theta}_\rho) - \nabla_{\theta_v} L(\theta_V, \theta_\pi, \theta_\rho^*)\| \leq \tilde{\varepsilon}_t$ and $\|\nabla_{\theta_\pi} L(\theta_V, \theta_\pi, \tilde{\theta}_\rho) - \nabla_{\theta_\pi} L(\theta_V, \theta_\pi, \theta_\rho^*)\| \leq \tilde{\varepsilon}_t$.

Before we begin the proof, we made some mild assumption on the function $f(x)$.

**Assumption 2.** 1. The function f(x) is differentiable and has Lipschitz continuous gradient, i.e.,
$$\|\nabla f(x) - \nabla f(y)\| \leq \|x - y\|, \forall x, y \in \mathbb{R}^K.$$

2. Further assume that $A^T A + B^T B \succeq I$. This assumption is always satisfied by our choice on A and B. We have $A^T A + B^T B \succeq D \succeq \min_i \{d_i\} I$

3. There exists a constant $\delta > 0$ such that $\exists \underline{f} > -\infty, s.t., f(x) + \frac{\delta}{2} \|Ax - b\|^2 \geq \underline{f}, \forall x$. This assumption is satisfied if we require the parameter space is bounded.

**Lemma 1.** *Suppose the assumption 2 is satisfied, we have following inequality holds*

$$\frac{\|\mu^{t+1} - \mu^t\|^2}{\beta} \leq \frac{3L^2}{\beta \sigma_{\min}} \|x^t - x^{t-1}\|^2 + \frac{3}{\beta} \|\epsilon_{t-1} - \epsilon_t\|^2 + \frac{3\beta}{\sigma_{\min}} \|B^T B ((x^{t+1} - x^t) - (x^t - x^{t-1}))\|^2 \qquad (17)$$

*Proof.* Using the optimality condition of (14), we obtain

$$\nabla f(x^t) + \epsilon^t + A^T \mu^t + \beta A^T (Ax^{t+1} - b) + \beta B^T B (x^{t+1} - x^t) = 0 \qquad (18)$$

applying equation (15) we have

$$A^T \mu^{t+1} = -\nabla f(x^t) - \beta B^T B (x^{t+1} - x^t). \qquad (19)$$

Note that from the fact that $\mu^0 = 0$, we have the variable lies in the column space of $A$.

$$\mu^r = \beta \sum_{t=1}^{r} (Ax^t - b).$$

510     Let $\sigma_{\min}$ denote the smallest non-zero eigenvalue of $A^T A$, we have

$$
\begin{aligned}
&\sigma_{\min}^{1/2}\|\mu^{t+1} - \mu^t\| \\
&\leq \|A(\mu^{t+1} - \mu^t)\| \\
&\leq \| -\nabla f(x^t) - \epsilon_t - \beta B^T B(x^{t+1} - x^t) - (-\nabla f(x^{t-1}) - \epsilon_{t-1} - \beta B^T B(x^t - x^{t-1}))\| \\
&= \|\nabla f(x^{t-1}) - \nabla f(x^t) + (\epsilon_{t-1} - \epsilon_t) - \beta B^T B((x^{t+1} - x^t - (x^t - x^{t-1})))\|.
\end{aligned}
\tag{20}
$$

511     Thus we have

$$
\begin{aligned}
&\frac{\|\mu^{t+1} - \mu^t\|^2}{\beta} \\
&\leq \frac{1}{\beta\sigma_{\min}^{1/2}} \|\nabla f(x^{t-1}) - \nabla f(x^t) + (\epsilon_{t-1} - \epsilon_t) - \beta B^T B((x^{t+1} - x^t - (x^t - x^{t-1})))\|^2 \\
&\leq \frac{3L^2}{\beta\sigma_{\min}} \|x^t - x^{t-1}\|^2 + \frac{3}{\beta} \|\epsilon_{t-1} - \epsilon_t\|^2 + \frac{3\beta}{\sigma_{\min}} \|B^T B((x^{t+1} - x^t) - (x^t - x^{t-1}))\|^2,
\end{aligned}
\tag{21}
$$

512     where the second inequality holds from the fact that $(a + b + c)^2 \leq 3a^2 + 3b^2 + 3c^2$.

513     $\qquad\qquad\qquad\qquad\qquad\qquad\qquad\qquad\qquad\qquad\qquad\qquad\qquad\qquad\qquad\qquad\qquad\qquad$ $\square$

514     **Lemma 2.** *Define $L_\beta(x^t, \mu^t) = f(x^t) + \langle \mu^t, Ax - b \rangle + \frac{\beta}{2}\|Ax - b\|^2 + \frac{\beta}{2}\|x - x^t\|_{B^T B}^2$. Suppose*
515     *assumptions are satisfied, then the following is true for the algorithm*

$$
\begin{aligned}
&L_\beta(x^{t+1}, \mu^{t+1}) - L_\beta(x^t, \mu^t) \\
&\leq -\frac{\beta}{2}\|x^{t+1} - x^t\|^2 + \frac{3L^2}{\beta\sigma_{\min}}\|x^t - x^{t-1}\|^2 + \frac{3}{\beta}\|\epsilon_{t-1} - \epsilon_t\|^2 + \frac{3\beta}{\sigma_{\min}}\|B^T B((x^{t+1} - x^t) - (x^t - x^{t-1}))\|^2 \\
&\quad + \langle \epsilon_t, x^{t+1} - x^t \rangle
\end{aligned}
\tag{22}
$$

516     *Proof.* By the Assumptions $A^T A + B^T B \geq I$, the objective function in (14) is strongly convex with
517     parameter $\beta$.

518     Using the optimality condition of $x^{t+1}$ and strong convexity, we have for any $x$,

$$
\begin{aligned}
&L_\beta(x, \mu^t) + \frac{\beta}{2}\|x - x^t\|_{B^T B}^2 - (L_\beta(x^{t+1}, \mu^t) + \frac{\beta}{2}\|x^{t+1} - x^t\|_{B^T B}^2) \\
&\geq \langle \nabla L_\beta(x^{t+1}, \mu^t) + \beta B^T B(x^{t+1} - x^t), x - x^{t+1} \rangle + \frac{\beta}{2}\|x^{t+1} - x\|^2
\end{aligned}
\tag{23}
$$

519 Now we start to provide a upper bound of $L_\beta(x^{t+1}, \mu^{t+1}) - L_\beta(x^t, \mu^t)$.

$$
\begin{aligned}
&L_\beta(x^{t+1}, \mu^{t+1}) - L_\beta(x^t, \mu^t) \\
=&L_\beta(x^{t+1}, \mu^{t+1}) - L_\beta(x^{t+1}, \mu^t) + L_\beta(x^{t+1}, \mu^t) - L_\beta(x^t, \mu^t) \\
\leq&L_\beta(x^{t+1}, \mu^{t+1}) - L_\beta(x^{t+1}, \mu^t) + L_\beta(x^{t+1}, \mu^t) + \frac{\beta}{2}\|x^{t+1} - x^t\|_{B^T B}^2 - L_\beta(x^t, \mu^t) \\
\overset{a}{\leq}&\frac{\|\mu^{t+1} - \mu^t\|}{\beta} + \langle \nabla L_\beta(x^{t+1}, \mu^t) + \beta B^T B(x^{t+1} - x^t), x^{t+1} - x^t\rangle - \frac{\beta}{2}\|x^{t+1} - x^t\|^2 \\
\overset{b}{\leq}&-\frac{\beta}{2}\|x^{t+1} - x^t\|^2 + \frac{\|\mu^{t+1} - \mu^t\|^2}{\beta} + \langle \epsilon_t, x^{t+1} - x^t\rangle \\
\overset{c}{\leq}&-\frac{\beta}{2}\|x^{t+1} - x^t\|^2 + \frac{3L^2}{\beta \sigma_{\min}}\|x^t - x^{t-1}\|^2 + \frac{3}{\beta}\|\epsilon_{t-1} - \epsilon_t\|^2 \\
&+ \frac{3\beta}{\sigma_{\min}}\|B^T B\big((x^{t+1} - x^t) - (x^t - x^{t-1})\big)\|^2 + \langle \epsilon_t, x^{t+1} - x^t\rangle,
\end{aligned}
\tag{24}
$$

520 where the inequality (a) holds from the update rule in (15) and a simple algebra from the expression
521 of $L_\beta(x, \mu)$. Inequality (b) comes from the optimality condition of (14). Particularly, we have

$$
g(x^t) + A^T \mu^t + \beta A^T(Ax - b) + \beta B^T B(x - x^t) = 0
$$

522 replace $g(x^t)$ by $\nabla f(x^t) + \epsilon_t$, we have the result. The inequality (c) holds using the Lemma 1.

523 $\qquad\qquad\qquad\qquad\qquad\qquad\qquad\qquad\qquad\qquad\qquad\qquad\qquad\qquad\qquad\qquad\qquad$ $\square$

524 **Lemma 3.** *Suppose Assumption 2 is satisfied, then the following condition holds.*

$$
\begin{aligned}
&\frac{\beta}{2}(\|Ax^{t+1} - b\|^2 + \|x^{t+1} - x^t\|_{B^T B}^2) \\
\leq&\frac{L}{2}\|x^{t+1} - x^t\|^2 + \frac{L}{2}\|x^t - x^{t-1}\|^2 + \frac{\beta}{2}(\|x^t - x^{t-1}\|_{B^T B}^2 + \|Ax^t - b\|^2) \\
&-\frac{\beta}{2}(\|(x^t - x^{t-1}) - (x^{t+1} - x^t)\|_{B^T B}^2 + \|A(x^{t+1} - x^t)\|^2) - \langle \epsilon_t - \epsilon_{t-1}, x^{t+1} - x^t\rangle
\end{aligned}
\tag{25}
$$

525 *Proof.* Using the optimality condition of $x^{t+1}$ and $x^t$ in the update rule in (14), we obtain

$$
\langle g(x^t) + A^T \mu^t + \beta A^T(Ax^{t+1} - b) + \beta B^T B(x^{t+1} - x^t), x^{t+1} - x\rangle \leq 0, \forall x
\tag{26}
$$

526 and

$$
\langle g(x^{t-1}) + A^T \mu^{t-1} + \beta A^T(Ax^t - b) + \beta B^T B(x^t - x^{t-1}), x^t - x\rangle \leq 0, \forall x
\tag{27}
$$

527 Replacing $g(x^t)$ by $\nabla f(x^t) + \epsilon_t$ and $g(x^{t-1})$ by $\nabla f(x^{t-1}) + \epsilon_{t-1}$, and using the update rule (15)

$$
\langle \nabla f(x^t) + \epsilon_t + A^T \mu^{t+1} + \beta B^T B(x^{t+1} - x^t), x^{t+1} - x\rangle \leq 0, \forall x
\tag{28}
$$

528

$$
\langle \nabla f(x^{t-1}) + \epsilon_{t-1} + A^T \mu^t + \beta B^T B(x^t - x^{t-1}), x^t - x\rangle \leq 0, \forall x
\tag{29}
$$

529 Now choose $x = x^t$ in the first inequality and $x = x^{t+1}$ in the second one, adding two inequalities
530 together, we obtain

$$
\langle \nabla f(x^t) - \nabla f(x^{t-1}) + \epsilon_t - \epsilon_{t-1} + A^T(\mu^{t+1} - \mu^t) + \beta B^T B\big((x^{t+1} - x^t) - (x^t - x^{t-1})\big), x^{t+1} - x^t\rangle \leq 0
\tag{30}
$$

531 Rearranging above terms, we have

$$
\begin{aligned}
&\langle A^T(\mu^{t+1} - \mu^t), x^{t+1} - x^t\rangle \\
\leq&-\langle \nabla f(x^t) - \nabla f(x^{t-1}) + \epsilon_t - \epsilon_{t-1} + \beta B^T B\big((x^{t+1} - x^t) - (x^t - x^{t-1})\big), x^{t+1} - x^t\rangle
\end{aligned}
\tag{31}
$$

We first re-express the lhs of above inequality.

$$
\begin{aligned}
&\langle A^T(\mu^{t+1} - \mu^t), x^{t+1} - x^t \rangle \\
=&\langle \beta A^T(Ax^{t+1} - b), x^{t+1} - x^t \rangle \\
=&\langle \beta(Ax^{t+1} - b), Ax^{t+1} - b - (Ax^t - b) \rangle \\
=&\beta\|Ax^{t+1} - b\|^2 - \beta\langle Ax^{t+1} - b, Ax^t - b \rangle \\
=&\frac{\beta}{2}(\|Ax^{t+1} - b\|^2 - \|Ax^t - b\|^2 + \|A(x^{t+1} - x^t)\|^2)
\end{aligned}
\tag{32}
$$

Next, we bound the rhs of (31).

$$
\begin{aligned}
&- \langle \nabla f(x^t) - \nabla f(x^{t-1}) + \epsilon_t - \epsilon_{t-1} + \beta B^T B\big((x^{t+1} - x^t) - (x^t - x^{t-1})\big), x^{t+1} - x^t \rangle \\
=&- \langle \nabla f(x^t) - \nabla f(x^{t-1}) + \epsilon_t - \epsilon_{t-1}, x^{t+1} - x^t \rangle - \beta\langle B^T B\big((x^{t+1} - x^t) - (x^t - x^{t-1})\big), x^{t+1} - x^t \rangle \\
\overset{a}{\leq}&\frac{L}{2}\|x^{t+1} - x^t\|^2 + \frac{1}{2L}\|\nabla f(x^t) - \nabla f(x^{t-1})\|^2 - \langle \epsilon_t - \epsilon_{t-1}, x^{t+1} - x^t \rangle \\
&- \beta\langle B^T B\big((x^{t+1} - x^t) - (x^t - x^{t-1})\big), x^{t+1} - x^t \rangle \\
\overset{b}{\leq}&\frac{L}{2}\|x^{t+1} - x^t\|^2 + \frac{L}{2}\|x^t - x^{t-1}\|^2 - \langle \epsilon_t - \epsilon_{t-1}, x^{t+1} - x^t \rangle \\
&- \beta\langle B^T B\big((x^{t+1} - x^t) - (x^t - x^{t-1})\big), x^{t+1} - x^t \rangle \\
=&\frac{L}{2}\|x^{t+1} - x^t\|^2 + \frac{L}{2}\|x^t - x^{t-1}\|^2 - \langle \epsilon_t - \epsilon_{t-1}, x^{t+1} - x^t \rangle \\
&+ \frac{\beta}{2}(\|x^t - x^{t-1}\|_{B^T B}^2 - \|x^{t+1} - x^t\|_{B^T B}^2 - \|(x^t - x^{t-1}) - (x^{t+1} - x^t)\|_{B^T B}^2),
\end{aligned}
\tag{33}
$$

where the inequality (a) uses Cauchy-Schwartz inequality, (b) holds from the smoothness assumption on $f$.

Combine all pieces together, we obtain

$$
\begin{aligned}
&\frac{\beta}{2}(\|Ax^{t+1} - b\|^2 + \|x^{t+1} - x^t\|_{B^T B}^2) \\
\leq&\frac{L}{2}\|x^{t+1} - x^t\|^2 + \frac{L}{2}\|x^t - x^{t-1}\|^2 + \frac{\beta}{2}(\|x^t - x^{t-1}\|_{B^T B}^2 + \|Ax^t - b\|^2) \\
&- \frac{\beta}{2}(\|(x^t - x^{t-1}) - (x^{t+1} - x^t)\|_{B^T B}^2 + \|A(x^{t+1} - x^t)\|^2) - \langle \epsilon_t - \epsilon_{t-1}, x^{t+1} - x^t \rangle
\end{aligned}
\tag{34}
$$

$\square$

Same with Hong et al. [2017], we define the potential function

$$
P_{c,\beta}(x^{t+1}, x^t, \mu^{t+1}) = L_\beta(x^{t+1}, \mu^{t+1}) + \frac{c\beta}{2}(\|Ax^{t+1} - b\|^2 + \|x^{t+1} - x^t\|_{B^T B}^2)
\tag{35}
$$

**Lemma 4.** *If Assumption 2 holds, we have following*

$$
\begin{aligned}
&P_{c,\beta}(x^{t+1}, x^t, \mu^{t+1}) \\
\leq&P_{c,\beta}(x^t, x^{t-1}, \mu^t) - \big(\frac{\beta}{2} - \frac{cL}{2} - \frac{2c+1}{2}\big)\|x^{t+1} - x^t\|^2 + \big(\frac{3L^2}{\beta\sigma_{\min}} + \frac{cL}{2}\big)\|x^{t-1} - x^t\|^2 \\
&- \big(\frac{c\beta}{2} - \frac{3\beta\|B^T B\|}{\sigma_{\min}}\big)\|(x^{t+1} - x^t) - (x^t - x^{t-1})\|_{B^T B}^2 + \big(\frac{c}{4} + \frac{\beta}{3}\big)\|\epsilon_{t-1} - \epsilon_t\|^2 + \frac{1}{2}\|\epsilon_t\|^2
\end{aligned}
\tag{36}
$$

*Proof.*

$$P_{c,\beta}(x^{t+1}, x^t, \mu^{t+1})$$

$$\leq L_\beta(x^t, \mu^t) + \frac{c\beta}{2}(\|x^t - x^{t-1}\|_{B^T B} + \|Ax^t - b\|^2) - (\frac{\beta}{2} - \frac{cL}{2})\|x^{t+1} - x^t\|^2$$

$$+ (\frac{3L^2}{\beta\sigma_{\min}} + \frac{cL}{2})\|x^{t-1} - x^t\|^2 - (\frac{c\beta}{2} - \frac{3\beta\|B^T B\|}{\sigma_{\min}})\|(x^{t+1} - x^t) - (x^t - x^{t-1})\|_{B^T B}^2$$

$$+ \frac{3}{\beta}\|\epsilon_t - \epsilon_{t-1}\|^2 + \langle \epsilon_t, x^{t+1} - x^t \rangle - c\langle \epsilon_t - \epsilon_{t-1}, x^{t+1} - x^t \rangle$$

$$\leq P_{c,\beta}(x^t, x^{t-1}, \mu^t) - (\frac{\beta}{2} - \frac{cL}{2})\|x^{t+1} - x^t\|^2 + (\frac{3L^2}{\beta\sigma_{\min}} + \frac{cL}{2})\|x^{t-1} - x^t\|^2$$

$$- (\frac{c\beta}{2} - \frac{3\beta\|B^T B\|}{\sigma_{\min}})\|(x^{t+1} - x^t) - (x^t - x^{t-1})\|_{B^T B}^2 + \frac{c}{4}\|\epsilon_{t-1} - \epsilon_t\|^2$$

$$+ c\|x^{t+1} - x^t\|^2 + \frac{1}{2}\|\epsilon_t\|^2 + \frac{1}{2}\|x^{t+1} - x^t\|^2 + \frac{\beta}{3}\|\epsilon_t - \epsilon_{t-1}\|^2$$

$$= P_{c\beta}(x^t, x^{t-1}, \mu^t) - (\frac{\beta}{2} - \frac{cL}{2} - \frac{2c+1}{2})\|x^{t+1} - x^t\|^2 + (\frac{3L^2}{\beta\sigma_{\min}} + \frac{cL}{2})\|x^{t-1} - x^t\|^2$$

$$- (\frac{c\beta}{2} - \frac{3\beta\|B^T B\|}{\sigma_{\min}})\|(x^{t+1} - x^t) - (x^t - x^{t-1})\|_{B^T B}^2 + (\frac{c}{4} + \frac{\beta}{3})\|\epsilon_{t-1} - \epsilon_t\|^2 + \frac{1}{2}\|\epsilon_t\|^2 \tag{37}$$

where the second inequality holds from the Cauchy-Schwartz inequality.

We require that

$$\frac{c\beta}{2} - \frac{3\beta\|B^T B\|}{\sigma_{\min}} \geq 0,$$

which is satisfied when

$$c \geq \frac{6\|B^T B\|}{\sigma_{\min}} \tag{38}$$

We further require

$$(\frac{\beta}{2} - \frac{cL}{2} - \frac{2c+1}{2}) \geq (\frac{3L^2}{\beta\sigma_{\min}} + \frac{cL}{2}),$$

which will be used later in the telescoping.

Thus we require

$$\beta \geq 2cL + 2c + 1 + \frac{6L^2}{\beta\sigma_{\min}}. \tag{39}$$

and choose $\beta \geq CL + \frac{2c+1}{2} + \frac{1}{2}\sqrt{(2cL + 2c + 1)^2 + \frac{24L^2}{\sigma_{\min}}}$ □

Now we do summation over both side of (36) and have

$$
\sum_{t=1}^{T}[(\frac{\beta}{2} - \frac{cL}{2} - \frac{2c+1}{2})\|x^{t+1} - x^t\|^2 - (\frac{3L^2}{\beta\sigma_{\min}} + \frac{cL}{2})\|x^{t-1} - x^t\|^2]
$$

$$
+ \sum_{t=1}^{T}(\frac{c\beta}{2} - \frac{3\beta\|B^T B\|}{\sigma_{\min}})\|(x^{t+1} - x^t) - (x^t - x^{t-1})\|_{B^T B}^2 \tag{40}
$$

$$
\leq P_{c\beta}(x^1, x^0, \mu^0) - P_{c\beta}(x^{T+1}, x^T, \mu^T) + \sum_{t=1}^{T}[(\frac{c}{4} + \frac{\beta}{3})\|\epsilon_{t-1} - \epsilon_t\|^2 + \frac{1}{2}\|\epsilon_t\|^2]
$$

rearrange terms of above inequality.

$$
\sum_{t=1}^{T-1}(\frac{\beta}{2} - \frac{cl}{2} - \frac{2c+1}{2} - \frac{3L^2}{\beta\sigma_{\min}} - \frac{cl}{2})\|x^{t+1} - x^t\|^2 + (\frac{\beta}{2} - \frac{cl}{2} - \frac{2c+1}{2})\|x^{T+1} - x^T\|^2
$$

$$
\leq P_{c\beta}(x^1, x^0, \mu^0) - P_{c\beta}(x^{T+1}, x^T, \mu^T) + (\frac{3L^2}{\beta\sigma_{\min}} + \frac{cl}{2})\|x^1 - x^0\|^2 + \sum_{t=1}^{T}[(\frac{c}{4} + \frac{\beta}{3})\|\epsilon_{t-1} - \epsilon_t\|^2 + \frac{1}{2}\|\epsilon_t\|^2]
$$

(41)

Next we show $P_{c\beta}$ is lower bounded

The following lemma is from Lemma 3.5 in Hong [2016], we present here for completeness.

**Lemma 5.** *Suppose Assumption 2 are satisfied, and $(c, \beta)$ are chosen according to (39) and (38). Then the following state holds true*

$$
\exists \underline{P} \text{ s.t., } P_{c\beta}(x^{t+1}, x^t, \mu^{t+1}) \geq \underline{P} > -\infty
$$

*Proof.*

$$
\begin{aligned}
L_\beta(x^{t+1}, \mu^{t+1}) &= f(x^{t+1}) + \langle \mu^{t+1}, Ax^{t+1} - b \rangle + \frac{\beta}{2}\|Ax^{t+1} - b\|^2 \\
&= f(x^{t+1}) + \frac{1}{\beta}\langle \mu^{t+1}, \mu^{t+1} - \mu^t \rangle + \frac{\beta}{2}\|Ax^{t+1} - b\|^2 \\
&= f(x^{t+1}) + \frac{1}{2\beta}(\|\mu^{t+1}\|^2 - \|\mu^t\|^2 + \|\mu^{t+1} - \mu^t\|^2) + \frac{\beta}{2}\|Ax^{t+1} - b\|^2.
\end{aligned}
\tag{42}
$$

Sum over both side, we obtain

$$
\sum_{t=1}^{T} L_\beta(x^{t+1}, \mu^{t+1}) = \sum_{t=1}^{T}\left(f(x^{t+1}) + \frac{\beta}{2}\|Ax^{t+1} - b\|^2 + \frac{1}{2\beta}\|\mu^{t+1} - \mu^t\|^2\right) + \frac{1}{2\beta}(\|\mu^{T+1}\|^2 - \|\mu^1\|^2)
\tag{43}
$$

By assumption 2, above sum is lower bounded, which implies that the sum of the potential function is also lower bounded (Recall $P_{c,\beta}(x^{t+1}, x^t, \mu^{t+1}) = L_\beta(x^{t+1}, \mu^{t+1}) + \frac{c\beta}{2}(\|Ax^{t+1} - b\|^2 + \|x^{t+1} - x^t\|_{B^T B}^2)$ ). Thus we have

$$
P_{c\beta}(x^{t+1}, x^t, \mu^{t+1}) > -\infty, \forall t > 0
$$

$\square$

In the next step, we are ready to provide the convergence rate. Following Hong [2016], we define the convergence criteria

$$
Q(x^{t+1}, \mu^{t+1}) = \|\nabla L_\beta(x^{t+1}, \mu^t)\|^2 + \|Ax^{t+1} - b\|^2
\tag{44}
$$

It is easy to see, when $Q(x^{t+1}, \mu^t) = 0$, $\nabla f(x) + A^T \mu = 0$ and $Ax = b$, which are KKT condition of the problem.

$$
\begin{aligned}
&\|\nabla L_\beta(x^t, \mu^{t-1})\|^2 \\
=&\|\nabla f(x^t) - \nabla f(x^{t-1}) + \epsilon_t - \epsilon_{t-1} + A^T(\mu^{t+1} - \mu^t) + \beta B^T B(x^{t+1} - x^t)\|^2 \\
\leq&4L^2\|x^t - x^{t-1}\|^2 + 4\|\mu^{t+1} - \mu^t\|^2\|A^T A\| + 4\beta^2\|B^T B(x^{t+1} - x^t)\|^2 + 4\|\epsilon_t - \epsilon_{t-1}\|^2
\end{aligned} \tag{45}
$$

Using the proof in Lemma 1, we know there exist two positive constants c1 c2 c3 c4

$$
Q(x^t, \mu^{t-1}) \leq c_1\|x^t - x^{t+1}\|^2 + c_2\|x^t - x^{t-1}\|^2 + c_3\|B^T B\big((x^{t+1} - x^t) - (x^t - x^{t-1})\big)\|^2 + c_4\|\epsilon_t - \epsilon_{t-1}\|^2.
$$

Using Lemma 4, we know there must exist a constant $\kappa$ such that

$$
\begin{aligned}
&\sum_{t=1}^{T-1} Q(x^t, \mu^{t-1}) \\
\leq&\kappa(P_{c\beta}(x^1, x^0, \mu^0) - P_{c\beta}(x^{T+1}, x^T, \mu^T)) + \sum_{t=1}^{T}[(\frac{c}{4} + \frac{\beta}{3})\|\epsilon_{t-1} - \epsilon_t\|^2 + \frac{1}{2}\|\epsilon_t\|^2]) + c_4\sum_{t=1}^{T-1}\|\epsilon_t - \epsilon_{t-1}\|^2 \\
\leq&\kappa(P_{c\beta}(x^1, x^0, \mu^0) - \underline{P}) + \sum_{t=1}^{T}[(\frac{c}{4} + \frac{\beta}{3})\|\epsilon_{t-1} - \epsilon_t\|^2 + \frac{1}{2}\|\epsilon_t\|^2]) + c_4\sum_{t=1}^{T-1}\|\epsilon_t - \epsilon_{t-1}\|^2
\end{aligned} \tag{46}
$$

Divide both side by $T$ and take expectation

$$
\begin{aligned}
\frac{1}{T}\mathbb{E}\sum_{t=1}^{T} Q(x^t, \mu^{t-1}) \leq& \frac{1}{T}\kappa(P_{c\beta}(x_1, x^0, \mu^0) - \underline{P}) + \frac{\kappa}{T}[\sum_{t=1}^{T}(\frac{c}{4} + \frac{\beta}{3})\mathbb{E}\|\epsilon_{t-1} - \epsilon_t\|^2 + \frac{1}{2}\|\epsilon_t\|^2] \\
&+ \frac{c_4}{T}\sum_{t=1}^{T-1}\mathbb{E}\|\epsilon_t - \epsilon_{t-1}\|^2
\end{aligned} \tag{47}
$$

Now we bound the R.H.S. of above equation.

Recall we choose the mini-batch size $\sqrt{T}$, $\epsilon_t = \varepsilon_t + \tilde{\varepsilon}_t$ and $\varepsilon_t \leq c_1/\sqrt{T}$

$$
\|\epsilon_{t-1} - \epsilon_t\|^2 \leq 2\mathbb{E}(\|\epsilon_{t-1}\|^2 + \|\epsilon_t\|^2) \leq 4\mathbb{E}(\|\varepsilon_t\|^2 + \|\tilde{\varepsilon}_t\|^2 + \|\varepsilon_{t-1}\|^2 + \|\tilde{\varepsilon}_{t-1}\|^2) \leq \frac{8c_1}{T} + \frac{8\sigma^2}{T} \tag{48}
$$

Similarly we can bound $\|\epsilon_t\|^2$. Combine all pieces together, we obtain

$$
\frac{1}{T}\mathbb{E}\sum_{t=1}^{T} Q(x^t, \mu^{t-1}) \leq \frac{1}{T}\kappa(P_{c\beta}(x_1, x^0, \mu^0) - \underline{P}) + \frac{1}{T}(\kappa c_5 + c_6\sigma^2),
$$

where $c_5, c_6$ are some universal positive constants.

Notice $\min_t \mathbb{E}Q(x^t, \mu^{t-1}) \leq \frac{1}{T}\mathbb{E}\sum_{t=1}^{T} Q(x^t, \mu^{t-1})$, we have $\min_t \mathbb{E}Q(x^t, \mu^{t-1}) \leq (C + \sigma^2)/T$ where $C$ is a universal positive constant.

## B.2 Convergence on the dual update

If the dual objective function is non-convex, we just follow the exact analysis in our proof on the primal problem. Notice the analysis on the dual update is easier than primal one, since we do not

574 have the error term $\tilde{\epsilon}_t$. Therefore, we have the algorithm converges to stationary solution with rate
575 $\mathcal{O}(1/T)$in criteria $Q$.

576 If the dual objective function is linear or convex, the update rule reduce to Extra [Hong, 2016, Shi
577 et al., 2015] the convergence result of stochastic setting can be adapted from the proof in [Shi et al.,
578 2015]. Since it is not the main contribution of this paper, we omit the proof here.