[Reviews · NeurIPS 2019]

Reviewer 1



High level comment: It is difficult to evaluate this paper since the contributions are quite diverse and sprinkled throughout the paper mixed with known results. It would be good to more clearly separate previous papers from the contributions made in this paper. In particular, I think it would be good to have a clearly separate method section. For example, currently 3.1 and 3.2 are under the (presumably) method section but most of it just repeats the derivation from Dai et al 2018. I think those parts belong into the background section and should not be stated as contributions. Detailed Comments: -This paper adds novel theoretical results applicable to the single-agent setting and also adds consensus optimization to extend SBEED to decentralized multi-agent settings. I think it would be much cleaner to separate out these contributions into different papers. This would also make it easier to follow which challenges are due to non-linear function approximators and which ones are due to the decentralized MARL setting. -It would be good to explain the setting more clearly in a background section. In particular, the "goal of the agents" is introduced in the method section (equation 3). Only here it becomes clear that the setting is fully cooperative, ie. each agent is trying to maximise the average reward across all agents. -There is a repeated claim about "exploiting the data more efficiently" due to being off-policy. It would be great to experimentally validate this claim. -The experiments would be much stronger if there were carried out on a more challenging testbed for multi-agent learning, rather than toy-tasks. I have not checked the proofs and maths in detail. grammar: -line 101: "..above formulation by introduce.." -> ..above formulation by introducing.. line 202: ".. snd denote the total training step T." -> ?? -line 128: "Apply the soft temporal .." -> "Applying the soft temporal .."

Reviewer 2



I am not an expert in decentralized optimization and its links to multi agent RL but the subject tackled in this paper is very interesting and propositions of the authors are relevant. The fact that their algorithm can use off-policy updates is for me a strong point and I got the feeling that the results presented in this paper extend current knowledge in an interesting and valuable way.

Reviewer 3



This paper tackles the problem of decentralized learning in multi-agent environments. While many recent approaches use a combination of centralized learning and decentralized execution, the decentralized learning paradigm is motivated by scenarios where a centralized agent (e.g. a value function) may be too expensive to use, or may have undesirable privacy implications. However, previous decentralized learning approaches haven’t been very effective for multi-agent problems. The paper proposes a new algorithm, value propagation, and prove that it converges in the non-linear function approximation case. To my knowledge, the value propagation algorithm is novel and interesting. I’m not aware of any other decentralized learning method with convergence guarantees in the non-linear setting. Thus, this is a solid contribution to the multi-agent learning literature. One thing that is not entirely clear to me though, is how the proposed value propagation algorithm enables the authors to prove convergence in the non-convex optimization setting. In other words, do existing decentralized deep multi-agent learning approaches (such as the MA-AC algorithm the authors compare to) fundamentally not converge? Or is it just that the authors have put more effort into finding a proof of convergence for their algorithm? Either way, the proof is interesting, but it would be nice to have some intuition as to why the primal-dual form used by the authors enables convergence. From a readability perspective, the introduction is well written. However, after that there are many grammatical errors (although the paper is still understandable). I’d recommend more detailed proofreading of the later sections for grammatical mistakes. One of my concerns about the paper writing is that there is very little intuition given for why the proposed algorithm works better than other decentralized learning approaches. Why is using the dual form the right thing to do? What makes it work better than the baselines considered? The authors briefly touch on the fact that value propagation is can be used off-policy (e.g. with a replay buffer), but I would like more details on this. My other concern about the paper is that the empirical support for the proposed algorithm isn’t very strong. The authors first show results on randomly sampled MDPs, however they do this only for ablation purposes, and don’t compare to any other decentralized learning method (the centralized method they compared to understandably performs better). The authors only compare their method to other decentralized learning approaches on one environment, which is the cooperative navigation task from the ‘particle world’ set of environments. They show that value propagation out-performs a baseline multi-agent actor-critic from (Zhang et al., 2018), and also a centralized learning algorithm with communication removed. I’d like for these plots to be shown for a longer number of episodes, since it seems like the baseline algorithms are still improving (so it’s unclear if value propagation works better, or just learns more quickly). It’d also be good to compare to some other standard decentralized learning methods without communication, such as independent Q-learning. Overall, this is a single, very simple environment, so it’s not clear to me whether these results will generalize to other domains. As such, I’d say the experimental results are on the lower end of ‘adequate’. I understand that it is a lot to ask of a paper to provide a novel algorithm, show a proof of convergence, have compelling intuition, and also have strong experimental results. I’m not saying the paper needs to have all of these ingredients. However, I think it’s important to either show: (1) strong empirical results, indicating that the method outperforms various baselines across a variety of tasks; or (2) that there is an intuitive reason for why the method works on some smaller set of tasks (ideally that is backed up by data / theory), so we can expect the method to work better in other environments; or (3) that the proposed method gives us some new scientific insight or understanding about the nature of the problem being tackled, even if it doesn’t give improved performance. In this current form, I don’t think the paper does any of these. Overall, this is a fairly good paper. Due to the issues mentioned above, I am giving it a borderline accept. I’d consider increasing my score if some of the issues mentioned above are addressed. Small comments: - L60: preserves the privacy -> preserves privacy - L101: by introduce -> by introducing - L102: it would be good to define what rho(s, a) is when introducing it here - Maybe introduce the networked multi-agent MDP in Section 2? - L120: we call our method fully-decentralized method -> we call our method fully-decentralized - L128: Apply the soft temporal consistency -> applying - L202: snd -> and - Formatting / spacing of Section 6.2 is strange --------------------------------------------- After rebuttal: I appreciate the effort the authors put into their rebuttal, including adding an additional experiment, giving some more explanation, and extending the lengths of existing graphs. I am updating my score from a 6 to a 7. However, I still have some concerns, which makes my true score closer to a '6.5': - It's unclear how simple this 'traffic light' game is. Thus, even with the addition of this result, the experimental support I'd say is still on the 'weak' side. - the author's answer to why their method is performing better than MA-AC (it is off-policy and therefore has better data efficiency) isn't explicitly supported with a separate experiment. While one could argue that the fact that their curve goes up faster (i.e. in a fixed smaller number of iterations it achieves higher reward) means that it is more data-efficient, this clearly isn't the only reason their method is better than MA-AC --- the curve doesn't just go up faster, it gets better *asymptotic* performance. - I have some concerns about whether the explanations given by the authors will be written into the paper in a way that's clear and understandable. - I also echo some of Reviewer 1's comments about restructuring the paper to separate the exposition of Dai et al. and of this paper's contributions. It's hard to know if this will be done in a satisfactory way unless the paper is resubmitted.

[Author Response · NeurIPS 2019]

We thank the reviewers for all constructive reviews and will correct all minor problems accordingly .

Reviewer 1: Thanks for the comments. **(1)** We add a new experiment on the traffic light control problem [4] according
to requirement of the reviewer. We compare value propagation with MA-AC, independent Q learning, PCL without
communication and value propagation with partial observation. See more details in the left panel of Figure 1. **(2)** We
introduced our setting (goal of our algorithm) in the introduction session (lines 39 to 43). We will highlight that in the
final version so that readers can understand it clearly. **(3)** We have tested our claim on data efficiency of the off-policy
method in the experiment. One baseline MA-AC is an on-policy algorithm, and our value propagation algorithm
outperforms that due to better data efficiency. **(4)** Thanks for the suggestion on the structure of the paper, we will rectify
it in the final version. **(5)** We summarized our main contribution in the contribution section (row 65-76).

Reviewer 2: Thanks for the comments. One possible real-application is the multi-agent autonomous driving system [1],
where each agent may not want to share its goal (reward), its driving policy and its evaluation on current condition
(value function) . However they still can cooperate to guarantee the safety of driving, e.g., giving way, through the local
communication induced by the communication graph.

Reviewer 3: Thanks for the comments. **(1)**. We run the cooperative navigation task with more episodes in the middle
(8 agents) and right panel (16 agents) of Figure 1 with an additional baseline independent Q learning. We add a new
experiment on the traffic light control in the left panel. **(2)**. The high-level explanation on the proof of convergence.
Most commonly used TD algorithms are **semi-gradient** algorithms [2], e.g., Q learning, SARSA, Q and V update in
actor-critic. When they optimize the squared TD-error, they do not calculate gradient w.r.t. the parameter $\theta$ of the
target (e.g. target in Q learning is $R + \gamma \max_a Q_\theta(s_{t+1}, a)$.). That is why they are so-called semi-gradient algorithm.
It would have the convergence problem combining with the function approximation, and off-policy learning (called
deadly triad in [2]). One way to survive is to use the **true gradient** method [2], such as Gradient TD (but it is just a
policy evaluation method). Value propagation is a true gradient method. It optimizes the objective function in row 100
(single agent case) and eq (8) in multi-agent setting. In optimization theory, even the objective function is non-convex,
we still can design some gradient-based algorithms which converge to the stationary point. **(3)** The reason to use
primal-dual form. If we directly optimize the primal error (e.g., the objective function in row 100), it would meet the
double sampling problem [3]. To get around this problem, value propagation introduces a dual variable. The high-level
idea to use the dual variable is similar to Gradient TD (see the w variable in [3]), but now we solve a much harder
control problem ( Gradient TD is designed for policy evaluation ). **(4)** Intuition on why value propagation is better
than MA-AC. MA-AC is on-policy, which requires new samples to be collected for each gradient step. This becomes
expensive, as the number of gradient steps and samples per step needed to learn an effective policy increases with task
complexity. Value propagation (off-policy) reuses past experience. In algorithm 1, to update dual or primal problem, we
randomly sample mini-batch of transition from the replay buffer. The reason that value propagation is off-policy is
proposition 1, which says for all pair (s,a), the temporal consistency holds.

Figure 1: Additional experiments. Left: traffic light control. There are 9 agents where each of them represents a traffic
light. Actions are phase transition of traffic light. Rewards are combinations of delays, waiting time, emergency stops of
vehicle. Middle and Right panel: cooperative navigation task with more episodes. We add a new baseline, independent
Q learning, in the experiment. Value propagation clearly outperforms MA-AC, PCL without communication, and
independent Q learning.

[1] Safe, Multi-Agent, Reinforcement Learning for Autonomous Driving. S Shalev-Shwartz et al. 2016

[2] Introduction to Reinforcement Learning with Function Approximation, Richard Sutton, Nips 2015 tutorial.

[3] Fast Gradient-Descent Methods for Temporal-Difference Learning with Linear Function Approximation. Richard
Sutton et al. ICML 2009

[4] Coordinated Deep Reinforcement Learners for Traffic Light Control. Elise van der Pol et al., Nips 2016 workshop


[Meta-Review · NeurIPS 2019]

All reviewers agree that this paper provide novel theoretical results applicable to the single-agent setting. However, the author should carefully reshape their paper according to the suggestions raised by the reviewers, particularly reviewer 1